materials science/electron microscopy/
synthetic chemistry

reduced graphene oxide, secondary ion batteries, anode materials, mesoporous

**Author for correspondence:**
Jaekook Kim
e-mail: jaekook@chonnam.ac.kr

This article has been edited by the Royal Society of Chemistry, including the commissioning, peer review process and editorial aspects up to the point of acceptance.
[†]Present address: Chemical Sciences and Engineering Division, Argonne National Laboratory, Lemont, IL, USA.
[‡]Present address: Bio/Energy Center, Korea Institute of Energy Research, Gwangju, Korea.

# Facile synthesis of reduced graphene oxide by modified Hummer's method as anode material for Li-, Na- and K-ion secondary batteries

Jeonggeun Jo[1], Seulgi Lee[1], Jihyeon Gim[1,†], Jinju Song[1,‡], Sungjin Kim[1], Vinod Mathew[1], Muhammad Hilmy Alfaruqi[1], Seokhun Kim[1], Jinsub Lim[2] and Jaekook Kim[1]

[1]Department of Materials Science and Engineering, Chonnam National University, 300 Yongbong-dong, Bukgu, Gwangju 61186, Republic of Korea
[2]Korea Institute of Industrial Technology (KITECH), Buk-gu, Gwangju 61012, South Korea

JK, 0000-0002-6638-249X

Reduced graphene oxide (rGO) sheets were synthesized by a modified Hummer's method without additional reducing procedures, such as chemical and thermal treatment, by appropriate drying of graphite oxide under ambient atmosphere. The use of a moderate drying temperature (250°C) led to mesoporous characteristics with enhanced electrochemical activity, as confirmed by electron microscopy and $N_2$ adsorption studies. The dimensions of the sheets ranged from nanometres to micrometres and these sheets were entangled with each other. These morphological features of rGO tend to facilitate the movement of guest ions larger than $Li^+$. Impressive electrochemical properties were achieved with the rGO electrodes using various charge-transfer ions, such as $Li^+$, $Na^+$ and $K^+$, along with high porosity. Notably, the feasibility of this system as the carbonaceous anode material for sodium battery systems is demonstrated. Furthermore, the results also suggest that the high-rate capability of the present rGO electrode can pave the way for improving the full cell characteristics, especially for preventing the potential drop in sodium-ion batteries and potassium-ion batteries, which are expected to replace the lithium-ion battery system

# 1. Introduction

Over the past decades, various carbonaceous graphite species, such as natural graphite, coke and graphitized carbon have been widely investigated as the anodes for rechargeable lithium-ion batteries (LIBs). Among the different types of carbon anodes, natural graphite appears to be the most promising candidate due to its high capacity, low cost and low electrode potential relative to lithium metal [1]. However, LIBs do not meet the performance criteria (including cost, charge/discharge rate, energy density and safety) needed to enter new markets such as those for all-electric vehicles and for the storage of electrical energy from renewable wind and/or solar plants [2]. In this respect, researchers have explored rechargeable battery systems with alternative charge carriers. In order to meet the increasing energy demand, the alternative charge carriers require being naturally abundant and cheap. Obviously, sodium-ion and potassium-ion batteries offer attractive features of low cost, lower toxicity, improved safety and the ability to use electrolytes with low decomposition potential [3].

The carbonaceous material, graphite is improper for batteries based on sodium-ion and potassium-ion systems because of the thermodynamic instability of the binary graphite-intercalated compound (GIC). Further, the interlayer distance of graphite (approx. 0.34 nm) is too narrow to intercalate sodium ions or potassium ions [4,5]. To overcome this, modifying the graphite materials by tuning their morphology [1], exploring co-intercalation phenomena [6] and using graphene or graphene oxide [7,8] have been reported.

In recent years, graphene and graphene oxide (GO) have been synthesized by various kinds of physical and chemical methods. Among these methods, chemical vapour deposition (CVD) and mechanical cleavage of graphite are the most widely used approaches for obtaining defect-free and high-quality graphene [9,10]. However, these techniques are highly time consuming and lead to a low production yield. In contrast to CVD, Hummer's method has been widely adopted for the synthesis of GO based on its ease and short execution time [11,12]. However, the surface of GO sheets prepared by Hummer's method is highly oxygenated, bearing hydroxyl, epoxide, diol, ketone and carboxyl functional groups that possess saturated sp3 carbon atoms linked to oxygen, which makes the GO sheets insulators [13]. GO can be converted to a semiconductor via chemical or thermal treatment to remove the oxygen-containing functional groups. This process allows GO to be reduced incrementally so that the electrical conductivity can be tuned over several orders of magnitude [14]. Moreover, the solvent molecules from the water washing/filtration step in Hummer's method were identified to remain in the interlayers of the hydrophilic as-prepared product. Upon subsequent annealing, the trapped molecules facilitate the formation of the porous morphology in the final product. Since porosity features are known to influence electrochemical properties, various solvents (other than water) like HBr, $NH_3$ and, recently, HCl with different vapour pressures were used to tune the porous formation of GO and thereby enhance their electrochemical performances. Specifically, the use of HCl with higher vapour pressure (than water) enabled to obtain porous morphology and simultaneously promote a partial reduction in the GO material during drying at 120°C for anode application in rechargeable LIBs [15].

Inspired by these works, the present work performed a systematic study on using concentrated HCl as the filtering solvent in a modified Hummer's method [13] to prepare GO at various drying temperatures of 120, 200, 250 and 300°C. No further chemical/thermal reduction procedure is followed. Electron microscopy and surface analyses confirmed the accordion morphology of rGO and their mesoporous characteristics. The effect of variation of the drying temperatures on the porous morphology and electrochemical properties in the rGO material was studied in detail. In other words, the feasibility of using the prepared rGO host for the insertion/de-insertion of various carrier ions (such as $Li^+$, $Na^+$, and $K^+$) for energy storage applications is demonstrated. As expected, our work confirmed that the drying temperatures also significantly influenced the surface and electrochemical properties. The present study thus showcases the possibility of using rGO as an electrode material for alternative energy storage systems.

# 2. Material and methods

## 2.1. Synthesis of rGO with an accordion morphology

In a typical procedure, graphene oxide was synthesized from artificial graphite using a modified Hummer's method. The typical procedure is as follows: graphite (2 g) and $NaNO_3$ (2 g) were combined with $H_2SO_4$ (90 ml) and stirred for 30 min in an ice bath. $KMnO_4$ (10 g) was added to the resulting solution, and the solution was then stirred at 50°C for 2 h. Deionized (DI) water (200 ml)

and $H_2O_2$ (12 ml, 35%) were then slowly added to the above solution, and the resulting solution was washed with HCl (300 ml, 10%). Additional washing with concentrated HCl (200 ml, 37%) afforded the GO product as a powder [15]. Subsequently, the GO powder samples were dried at 120, 200, 250 or 300°C for 2 h to generate rGO that were exfoliated. The heating rate used for preparing the samples was 5°C min$^{-1}$.

## 2.2. Structure and morphology characterization

The powder X-ray diffraction pattern (PXRD) of the prepared sample was recorded using a Shimadzu X-ray diffractometer with Ni-filtered Cu K$\alpha$ radiation ($\lambda = 1.5406$ Å) operating at 40 kV and 30 mA at $2\theta$ values ranging from 10° to 80° in steps of 0.02°. The particle morphologies and sizes were determined using high-resolution transmission electron microscopy (HR-TEM). The TEM images were recorded using an FEI Tecnai$^{TM}$ F20 (at the Korea Basic Science Institute (KBSI), Gwangju Center) at 200 kV, and field emission scanning electron microscopy (FE-SEM) measurements were performed using a HITACHI S-4700 instrument. The degree of oxidation and the content of functional groups in the synthesized samples were analysed using an elemental analyser (EA, Vario-Micro Cube; KBSI, Busan, South Korea). The expanded graphene oxide samples were studied by using a LabRam HR800 UV Raman microscope (Horiba Jobin-Yvon, France; KBSI, Gwangju, Chonnam National University, South Korea) using 514 nm (10 mW) laser excitation. The porous features (including specific surface area and pore diameter distributions) of the final products were analysed by using the Brunauer–Emmett–Teller (BET) and Barrett–Joyner–Halenda (BJH) methods from the $N_2$ adsorption studies performed on an ASAP 2020 apparatus (Micromeritics Instrument Co., Norcross, GA, USA). The degree of oxidation and the content of functional groups in the synthesized samples were studied by X-ray photoelectron spectroscopy (XPS) using a Thermo VG Scientific instrument (Multilab 2000).

## 2.3. Electrochemical characterization

The electrodes were fabricated using the active material (AM), Super P and polyvinylidene fluoride (PVDF) binder in $N$-methyl-2-pyrrolidone (NMP) with a weight ratio of 80 : 10 : 10, respectively. The slurry was uniformly applied onto a Cu foil, to be used as the current collector, using the doctor blade method, dried at 80°C in a vacuum oven and then pressed between stainless steel twin rollers. The foil was then punched into circular discs and dried before assembling a cell. Coin cells were then assembled using lithium metal, sodium metal and potassium metal, respectively, as the counter electrode and a glass filter as the separator. Solutions of 1.0 M LiPF$_6$, 1.0 M NaClO$_4$ and 1.0 M KPF$_6$, respectively, in ethylene carbonate/dimethyl carbonate (EC/DMC, 1 : 1 by volume), propylene carbonate (PC), ethylene carbonate/ethylmethyl carbonate (EC/EMC 1 : 1 by volume) were used as electrolytes for the lithium, sodium and potassium test cells. Each of these 2032 coin-type cells was fabricated in an Ar-filled glove box and aged for 12 h before the electrochemical test. Galvanostatic studies, potentiostatic electrochemical impedance spectroscopy (PEIS) and cyclic voltammetry studies were performed on the coin cells fabricated from the prepared electrode samples using a programmable battery tester (BTS-2004H, Nagano, Japan and Bio-Logic Science Instruments) at 0.02–2.0 V under various current densities.

## 3. Results

FE-SEM images of rGO formed by the modified Hummer's method with increasing drying temperatures (dried at 120, 200, 250 or 300°C) are shown in figure 1 (hereafter mentioned as rGO-120, -200, -250 and -300, respectively). The images reveal the structurally exfoliated two-dimensional sheets and also show the tendency of steadily increasing sheet dimensions and porosity with temperature. In more detail, the rGO sheet dimensions ranging from nanometres to micrometres appear to be entangled with each other. This morphology of rGO could provide a high specific surface area which is suitable for the transport of larger ions such as sodium and potassium. Overall, the SEM images reveal a slight aggregation of the layers in the prepared samples, especially for the lower temperature samples. Furthermore, the TEM image of the sample prepared at 250°C in figure 1f details the morphology and porous features of the rGO. The Raman spectra and X-ray diffraction patterns for the rGO samples prepared by the modified Hummer's method with increasing drying temperatures are shown in figure 2a,b, respectively. Figure 2a shows that the G-band (approx. 1595.0 cm$^{-1}$ corresponding to sp2-

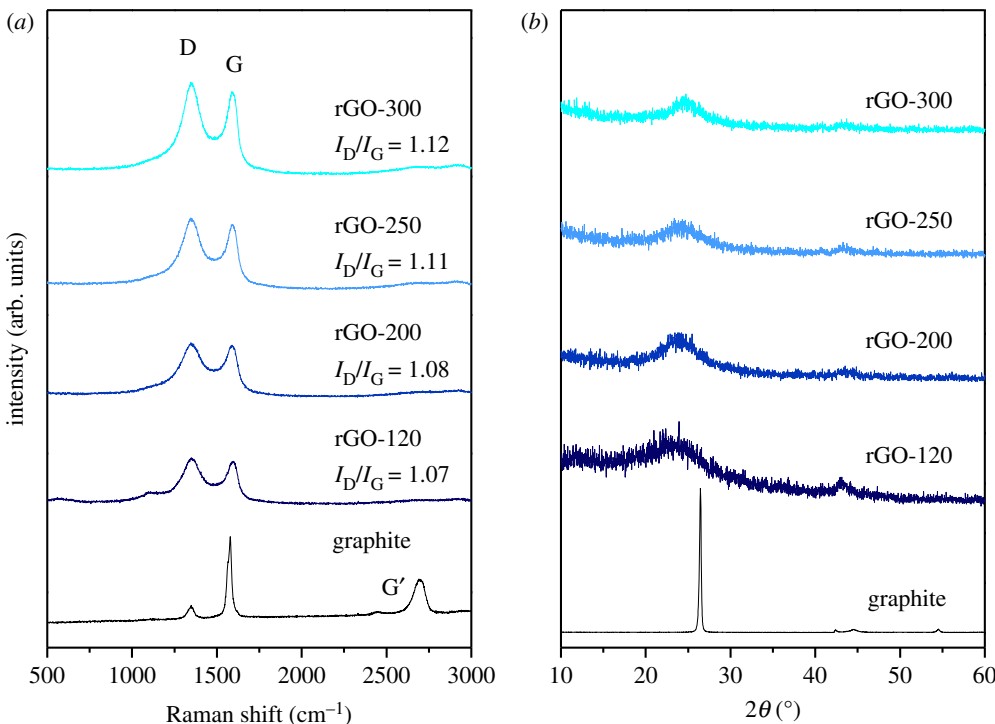

**Figure 1.** SEM images of (a) pure graphite, (b) rGO-120, (c) rGO-200, (d) rGO-250, (e) rGO-300 and HR-TEM image of (f) rGO-250 prepared by the modified Hummer's method.

**Figure 2.** RAMAN spectra and XRD patterns of the rGO samples synthesized at various drying temperatures compared to the corresponding spectra of graphite.

hybridized carbon) and the D-band (approx. 1353.5 cm$^{-1}$) coincide for all of the rGO samples prepared by the modified Hummer's method. This observation confirms that the disordered carbon in the composite corresponds to the carbon in rGO. Moreover, the observed bands match well with those reported in the literature [16]. Generally, the intensity of the D-band is related to the size of the in-plane sp2 domains [17]. An increasing D peak intensity indicates the formation of more sp2 domains

**Table 1.** Elemental analysis result of rGO samples synthesized at various drying temperatures with the corresponding data of graphite as reference.

| | carbon content (wt%) | oxygen content (wt%) | C/O ratio |
|---|---|---|---|
| graphite | 99.57 | — | — |
| rGO-120 | 33.61 | 49.46 | 0.679 |
| rGO-200 | 66.19 | 25.77 | 2.587 |
| rGO-250 | 70.54 | 22.81 | 3.095 |
| rGO-300 | 70.74 | 24.46 | 2.89 |

while the relative intensity ratio between the characteristic peaks ($I_D/I_G$) indicates the degree of disorder and is inversely proportional to the average size of the sp2 clusters [17,18]. As shown in figure 2a, the $I_D/I_G$ for the rGO samples tends to slightly increase with increasing drying temperatures. These results suggest that new (or more) graphitic domains were formed and the sp2 cluster number increased [19] with the drying temperature thus demonstrating that good reduction efficiency is influenced by the applied temperature for drying. Figure 2b shows the XRD patterns of graphite and rGO samples dried at different temperatures. As expected, the XRD profile of graphite shows an intense peak at 26.5° and that of rGO samples shows a broad peak around 24° [20]. Notably, the broad peaks of rGO samples are similar in shape, whereas the peak position is shifted to higher $2\theta$ value with increasing drying temperature and hence suggests a more extensive reduction in the rGO samples prepared at moderate drying temperatures [21]. The XRD data thus clearly indicate the influence of the thermal treatment on the reduction of graphene oxide.

To confirm the decomposition of oxygen with increasing drying temperature, elemental analysis was conducted and the corresponding results are shown in table 1. As the drying temperature increased from 120 to 250°C, the oxygen content of rGO declined dramatically. In other words, the C/O intensity ratio estimation from the elemental analysis or the XPS data gives a measure of the degree of reduction in the prepared samples and these results are congruent with each other [22]. A higher de-oxygenation process occurs due to the reduction process compared to that in the oxidized (less reduced) samples. Accordingly, the C/O ratio increased for increasing drying temperatures and is the maximum (3.095) at 250°C drying temperature, as observed from table 1. Interestingly, the oxygen content is slightly higher for the rGO prepared at 300°C than that prepared at 250°C. This can be related to the open-air environments used for the drying process in the present study as the annealing environment can influence the thermal reduction of GO. Usually, controlled reaction environments like inert air atmosphere or vacuum conditions are followed for the thermal reduction of GO. In the present case, the surface functional groups containing oxygen (absorbed water and hydroxyl/carboxyl group) are decomposed at temperatures under 250°C [22]. Given that the open-air conditions are oxidizing atmospheres, there can be a slight increase in oxygen content via some chemical reactions like the chemisorption of oxygen by the active surface carbon. Also, it is possible that the impurities present in the open air could interfere with the sample and cause undesired reactions. Hence, more studies are required to identify the exact reasons for the slight increase in the oxygen content at 300°C. However, the overall trend clearly indicates that GO undergoes thermal reduction upon drying beyond the boiling point of hydrochloric acid [23]. For further confirmation, XPS studies of rGO samples dried at various temperatures were performed and the resultant spectra indicating the chemical states of the constituent elements are presented in figure 3. For all the prepared samples, the C1s spectrum could be deconvoluted into three peaks at 284.5, 286.0 and 287.9 eV (figure 3a), corresponding to different carbon functionalities such as the non-oxygenated ring C (C–C), the C bound to O (C–O), and carbonyl C (C = O), respectively. The carboxylate carbon (O–C = O) at 289.0 eV, however, could not be traced by the deconvolution of the C1s spectrum. Although all three peaks were observed in all four samples, the intensities of particularly the oxygenated bonds became weaker at higher drying temperatures. For example, after drying at moderate temperatures (greater than 120°C), the intensity of the C–O signal decreased, whereas the intensity of the C–C signal increased. Specifically, the estimated C/O intensity ratios were 0.66, 1.5, 1.98 and 1.8 for the graphene oxide samples prepared at drying temperatures of 120, 200, 250 and 300°C, respectively, the highest value observed for the rGO dried at 250°C. As expected, the estimated C/O intensity ratio values calculated by the XPS results are mostly consistent with those of the elemental analysis due to the fact

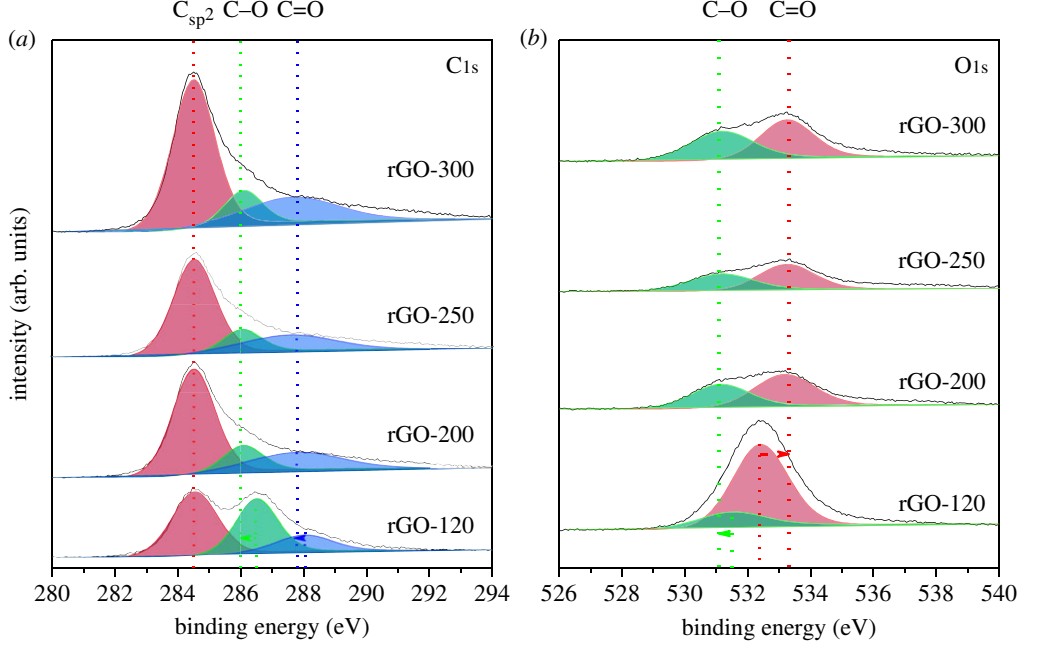

**Figure 3.** XPS spectra of rGO samples synthesized at various drying temperatures. (*a*) Deconvolution of C1s, (*b*) deconvolution of O1s.

**Table 2.** BET analysis results (indicating surface area and pore volume) of rGO samples synthesized at various drying temperatures with the corresponding data of graphite as reference.

|  | BET surface area (m$^2$ g$^{-1}$) | pore volume (cm$^3$ g$^{-1}$) |
|---|---|---|
| graphite | 1.0293 | 0.0075 |
| rGO-120 | 3.5581 | 0.0244 |
| rGO-200 | 56.416 | 0.3088 |
| rGO-250 | 148.89 | 0.812 |
| rGO-300 | 148.32 | 0.7973 |

that the former analysis represents the surface composition while the latter measurement corresponds to the bulk of the material [22]. Also, since the drying temperatures have been maintained below 500°C, the estimated C/O values are comparatively less than those reported [24]. Furthermore, the O1s spectrum could be deconvoluted into two peaks at 531.1 and 533.3 eV (figure 3*b*), which correspond to the presence of different oxygen functionalities, such as C = O and C–O, respectively. Deconvolution of the O1s spectra was performed by considering two different contributions due to oxygen atoms that are single- and double-bonded to one carbon atom [25,26].

At a moderate temperature of 250°C, the intensity of the O1s signal corresponding to C–O is apparently less compared to the spectra recorded for lower temperatures; this trend being similar when the corresponding C1s XPS spectra are compared [27]. The observation of such similarities in the C1s and O1s spectra confirm that both oxidation and reduction take place concurrently: hydroxyl and epoxy groups are progressively eliminated, and newly formed oxidation groups simultaneously appear. Thus, the final product assumes a partly restored graphene sp2 network bearing oxygen-containing groups and this combination tends to make the rGO electrically conductive [28]. Therefore, the Raman, elemental and XPS analyses clearly demonstrate that GO is easily reduced under ambient atmosphere and also tend to suggest an enhancement in the electrical conduction characteristics and energy storage abilities with the formation of rGO [28,29].

BET and BJH analyses were performed under N$_2$ gas to understand the porous features of the prepared rGO samples and the corresponding results are summarized in table 2 and figure 4. At the drying temperature of 120°C, which is close to the boiling point of water, the specific surface area is

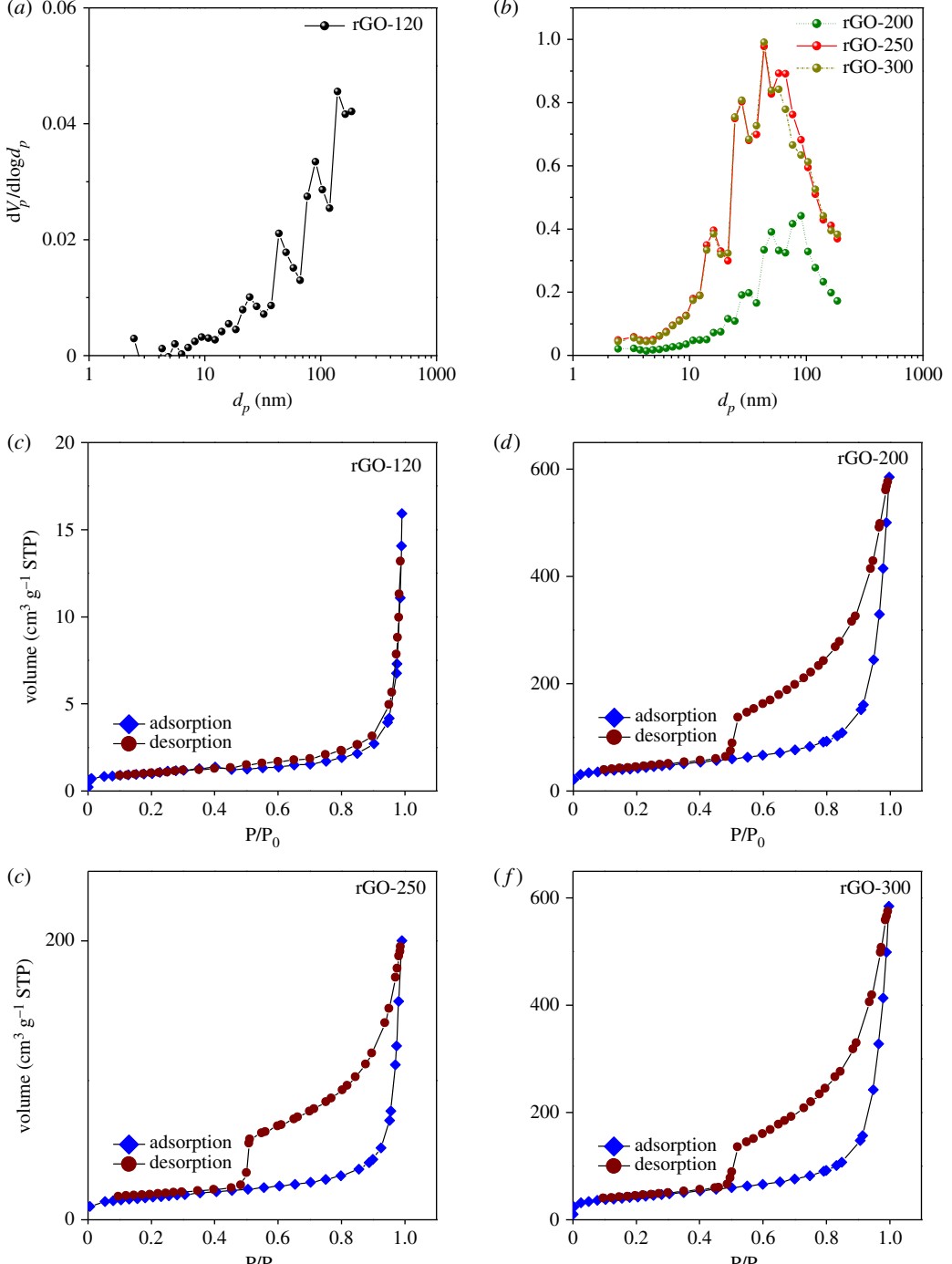

**Figure 4.** (a,b) BJH analysis results (indicating pore size and distribution) and (c–f) N$_2$ adsorption isotherms for the rGO samples synthesized at various drying temperatures in the present study.

similar to that of pristine graphite. The corresponding BJH plot in figure 4a reveals a wide distribution of hierarchical pores with diameters ranging from the meso to macro range. And the corresponding adsorption/desorption plot reveals a type II isotherm with H3 hysteresis loop. This behaviour is more related to non-rigid aggregates of plate-like particles with slit-like pores in the rGO prepared at 120°C drying temperature [30]. However, when the drying temperature reached 200°C, wherein decomposition of the oxygen-containing functional groups begins, the specific surface area and pore volume increased dramatically from 1.02 to 56.41 m$^2$ g$^{-1}$ and 0.0075 to 0.3088 cm$^3$ g$^{-1}$, respectively. Similar to the case of the sample dried at 120°C, the BJH plot revealed hierarchical pores except that the macroporous range of pore distribution is reduced by one order of magnitude. This may be

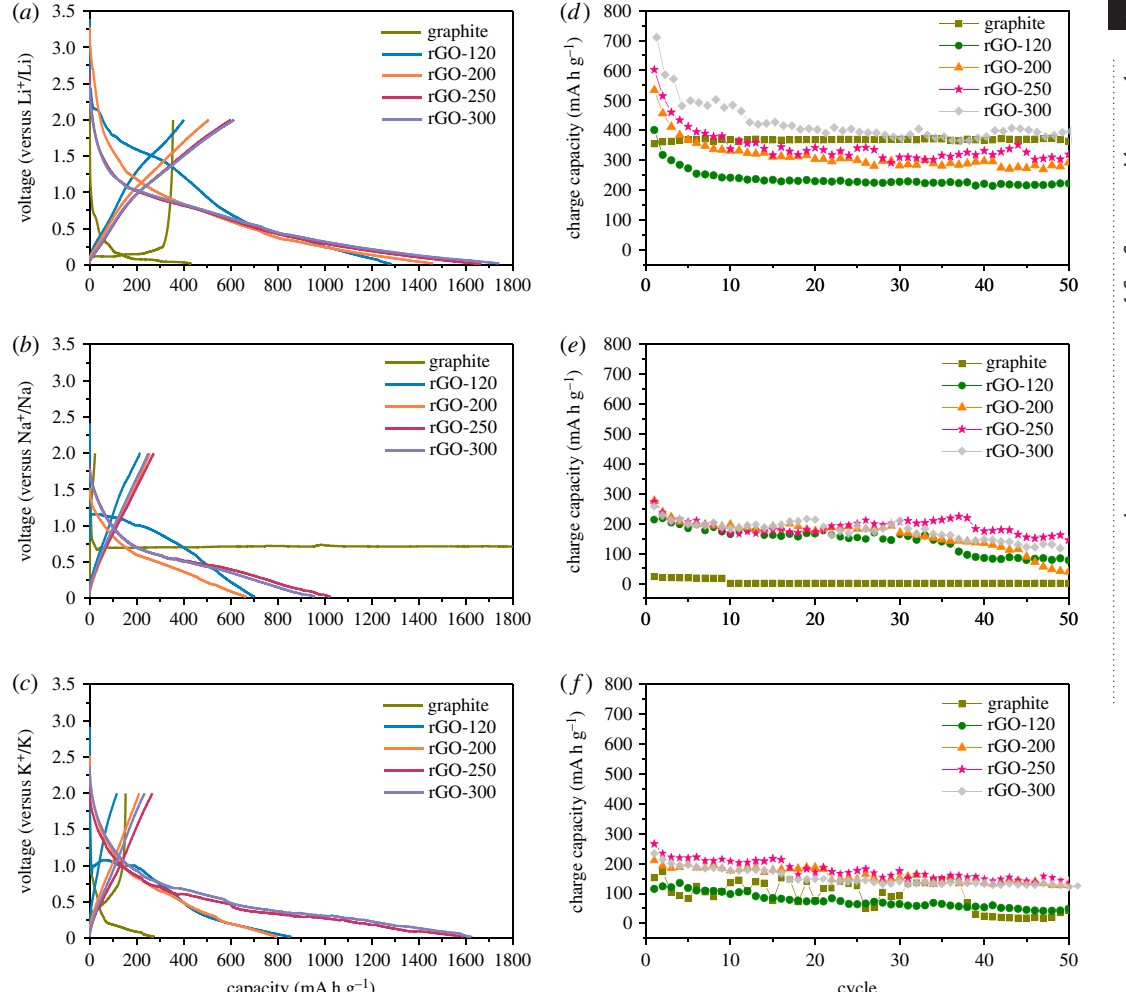

**Figure 5.** First discharge/charge curves and cyclability of graphite and rGO samples synthesized at various drying temperature in (a,d) lithium, (b,e) sodium and (c,f) potassium half-cells at 0.1 C current rate in the voltage range 0.02−2.0 V.

related to the shrinkage of pores occurring at higher drying temperatures. Notably, at 250°C, the specific surface area and pore volume further increased and reached their maximum values of 148.89 m$^2$ g$^{-1}$ (an increase by one order of magnitude) and 0.8122 cm$^3$ g$^{-1}$, respectively. Correspondingly, the pore-size distribution plot clearly indicates the presence of just hierarchical mesopores, thereby suggesting further shrinkage in the pore diameters compared to that observed for the 200°C dried sample. On exceeding 250°C, the specific surface area, pore volume and pore size did not vary much compared to that of rGO dried at 250°C, as is evident from table 2 and figure 4b. The trend of the maximum surface variations occurring at 250°C is due to the exothermic reaction related to the elimination of physisorbed and interlamellar water molecules [31]. The maximum surface area values obtained here are competitive with those reported for the case of rGO prepared for various applications [8,32–34].

Furthermore, figure 4c−f indicates that the pore-size distribution in the mesopore range for the sample optimized at 250°C drying temperature is highly advantageous for electrochemical ion insertion. Similar to the case of the sample dried at 120°C, the N$_2$ adsorption curves reveal a type II isotherm with H3 hysteresis for all other samples thus revealing the porous characteristics. Overall, the electron microscopy, Raman, XPS and BET/BJH results support each other and clearly demonstrate that the drying temperature applied for the rGO prepared using the modified Hummer's method influences their morphological and porous features. More importantly, these studies clearly confirm that the optimized drying temperature for producing rGO with mesoporous characteristics favouring large guest-ion insertion/de-insertion is 250°C.

Electrochemical properties of the prepared rGO samples were evaluated at 0.1 C (37.5 mA g$^{-1}$) in the voltage range of 0.02−2.0 V in order to measure their storage abilities of lithium ion, sodium ion and potassium ion, respectively, and the resultant initial voltage profiles are shown in figure 5a−c. All the

prepared samples displayed a largely irreversible plateau around 0.5 V, which can be attributed to the formation of a solid electrolyte interphase (SEI) film on the electrode surface [35]. Despite this, reversible capacities in the ranges of 400–600, 200–250 and 100–200 mA h g$^{-1}$ are achievable for the rGO electrodes used in half-cells of lithium, sodium and potassium, respectively. Apparently, the charge capacity for all samples increases with increasing drying temperature for all the cases of lithium, sodium and potassium, respectively. The cyclic voltammetry profiles recorded for the lithium test cells (electronic supplementary material, figure S1) clearly support the trend of the voltage profiles showing increased specific capacity for higher drying temperatures. This result is related to changes in the surface area, porous features such as pore size/volume and interlayer distance compared with that of graphite due to the reduction process, which leads to a relatively large storage capacity for various transport ions [36–38]. Notably, when used as an anode material for sodium-ion batteries, graphite (carbonaceous material) facilitates inappreciable sodium-ion intercalation. Moreover, the binary Na-intercalated GICs undergo electrochemically irreversible reactions and are thermodynamically unstable [39,40]. However, in this report, we demonstrate the feasibility of using rGO anode material for the sodium battery system. This is related to the fact that the energy storage mechanism is surface-driven, wherein the reaction between sodium ion and the defects on the surface of the rGOs advantageously influences the interfacial activity between the electrode and electrolyte [41]. Overall, the results in figure 5 demonstrate that the rGO-250 sample has impressive lithium, sodium and potassium storage properties. Precisely, the rGO-250 electrode displayed high initial reversible capacities of 602, 272 and 266 mA h g$^{-1}$ when used in lithium, sodium and potassium half-cells, respectively. For example, Wang *et al.* [8] developed porous rGO with high surface area (approx. 330 m$^2$ g$^{-1}$) and the electrochemical measurement revealed reversible sodium storage capacities of 174 mA h g$^{-1}$ at 0.2C. Although the surface area is higher than that measured in the present case, the reversible specific capacity attained here is quite competitive with the value reported. The general decrease in the specific capacities during the initial few cycles of the present samples can arise from the stabilization of the SEI layer on the electrode surface [8]. However, the slight instability in the specific capacities during repeated cycling can be related to the slightly aggregated layers in the prepared electrode samples, and further investigations towards layer exfoliation in the rGO samples via chemical or mechanical or thermal methods are required. Interestingly, the irreversible capacities of rGO-250 in the sodium and potassium test cells are lower than that observed for the corresponding lithium test cell. This is related to the difference in their standard electrode potentials (Li < Na < K), which is affected by SEI layer formation. In the present case, the SEI layer was formed around 0.5 V and the specific capacity is related to the chemical formula of the fully guest-ion intercalated rGO obtained at the discharge cut-off potential at 0.02 V. Upon complete discharge cycling of the lithium, sodium and potassium test cells, respectively, the intercalated rGO product corresponds to LiC$_6$, Na$_\delta$C$_{64}$ ($\delta \sim 1$) and KC$_8$ [42,43].

The slightly better electrochemical performance of the rGO sample prepared at 250°C, especially for sodium and potassium test cells, requires to be reasoned out. Therefore, to clearly understand the influence of the drying temperature toward the electrochemical performance of rGO electrode in coin cell, PEIS measurement was performed for all rGO samples with Na metal anode under a determined voltage range of 0.02–2.0 V and 0.5 C (187.5 mA g$^{-1}$) current density. Figure 6a represents the Nyquist plot of the first-charge impedance behaviour in the frequency range of 1 MHz–10 mHz. The compressed semicircle magnitude is steadily enlarged with raising temperatures and ended at 28.3, 38.7 and 48.9 Ω for the rGO-120, -200 and -300 samples, respectively. Interestingly, the rGO-250 sample exhibited an out-of-trend behaviour thereby suggesting a different mechanism. Hence, an electrical equivalent circuit (EEC) analysis was performed using Z-VIEW software, providing a closest analogue description of the electrochemical kinetics. Accordingly, the EEC contained a resistor $R_s$ in series with two parallel RC combinations ($R_{SEI}$ + CPE1, $R_{CT}$ + CPE2) and connected to a Warburg element, $W_{diffusion}$, as shown in figure 6a (inset). The detailed information of these components is listed in table 3. According to our previous study, the physical meaning of the EEC elements that describe the steps for lithium-ion insertion are proposed as follows: (i) $R_s$ is attributed to the ion transport inside the separator and the electrolyte; (ii) the $R_{SEI}$ + CPE1 combination represents lithium-ion migration through the SEI layer; (iii) the $R_{CT}$ + CPE2 group reflects the charge-transfer at the electrode/electrolyte interface; and (iv) $W_{diffusion}$ illustrates the lithium-ion diffusion inside the bulk phase of the AM [44,45]. Taking a cue from these, the variations of $R_S$, $R_{SEI}$, $R_{CT}$ and $R_{diffusion}$ values at the first-charge state regarding different temperature samples for the sodium test cells are provided in figure 6b. The system ($R_S$) and SEI layer ($R_{SEI}$) resistances occupy a minor distribution in the total resistance, which is steadily retained for all samples. Furthermore, the charge-transfer resistance ($R_{CT}$)

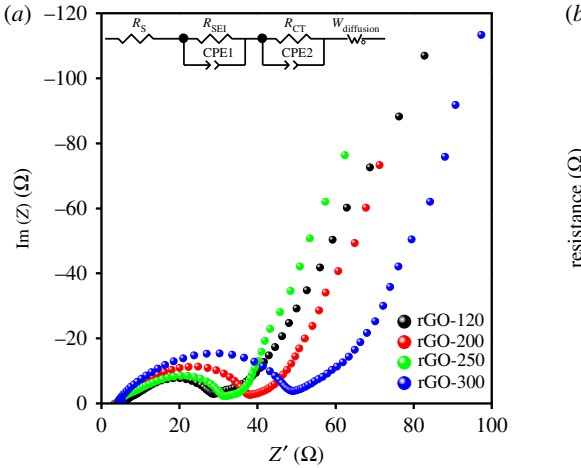
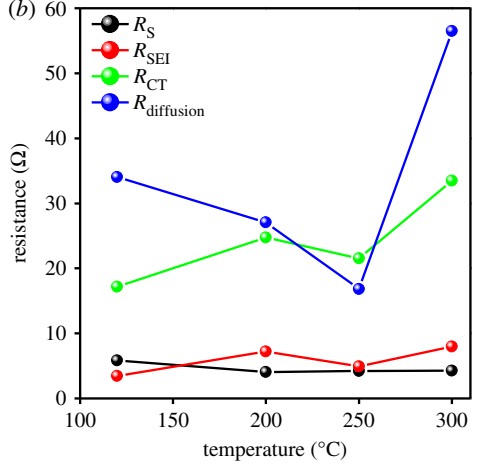

**Figure 6.** The (*a*) Nyquist plots and the equivalent circuit (inset) and (*b*) the different fitted resistance values for the prepared rGO samples in sodium test cells after one charge cycle.

**Table 3.** Fitting values of equivalent circuit components in *in situ* PEIS measurement performed in the first-charge state.

|  | $R_S$ | $R_{SE1}$ | $R_{CT}$ | $R_{diffusion}$ |
|---|---|---|---|---|
| rGO-120 | 5.832 | 3.457 | 17.19 | 34.06 |
| rGO-200 | 4.045 | 7.212 | 24.75 | 27.06 |
| rGO-250 | 4.211 | 4.912 | 21.54 | 16.81 |
| rGO-300 | 4.246 | 7.996 | 33.46 | 56.49 |

tends to increase at the higher temperature of 200 and 300°C samples. Most importantly, the diffusion resistance ($R_{diffusion}$) of rGO-250 sample shows the lowest values of 16.8 Ω more than three-fold less than rGO-300 samples (56.49 Ω). This implies a drastic improvement in the ion-diffusion conductivity due to the optimized drying temperature of 250°C. The PEIS results, therefore, match well with the galvanostatic results discussed earlier. Additionally, the best performance is related to their high-specific surface area and optimum pore-size distribution, as indicated in table 2. Furthermore, the defects on the surface of rGO influence the storage of the transport ions.

In order to understand the more detailed electrochemical properties of rGO-250 sample, the cycling and C-rate performances were evaluated and the results are presented in figure 7. The cycle stability and coulombic efficiency plots for the rGO-250 electrodes used in lithium, sodium and potassium test half-cells, respectively, are shown in figure 7*a*. After 50 cycles, the reversible specific capacity of the Li/Na half-cell remained at 53% while that of the K half-cell remained at 51% compared with their corresponding first charge capacities. Except the initial few cycles, the coulombic efficiency remained close to 100% for all the test cells during the entire cycling process. This result indicates appreciable cycling stabilities of rGO anodes with respect to lithium, sodium and potassium storage. The rate performances of rGO-250 with respect to lithium, sodium and potassium are shown in figure 7*b*. Remarkably, at a high rate of 12.8 C (480 mA g$^{-1}$), 20%, 29% and 31% of initial specific capacities (at 0.1 C) are obtained for the lithium, sodium and potassium test cells, respectively. Further, the capacity recovery performance was almost 70%, except for the lithium test cell. This result confirms that the electrode operates well at high current densities and also shows superior capacity recoverability when returning from a high value of 12.8 C to a low current density of 0.1 C. Although the aspect of gradual specific capacity decrease, in general, under repeated cycling is most likely related to the aggregated layers in the rGO samples, there is room for further improvement in the electrochemical properties using simple strategies like solvent/surfactant-assisted ultrasonication and/or mechanical/thermal methods. In addition, the possibility of tuning other parameters including the annealing environment and heating rate could also enable the realization of enhanced electrochemical properties.

The present work, therefore, reported on the synthesis of rGO by a modified Hummer's method using HCl with higher vapour pressure (than water) as the filtration solvent followed by subsequent

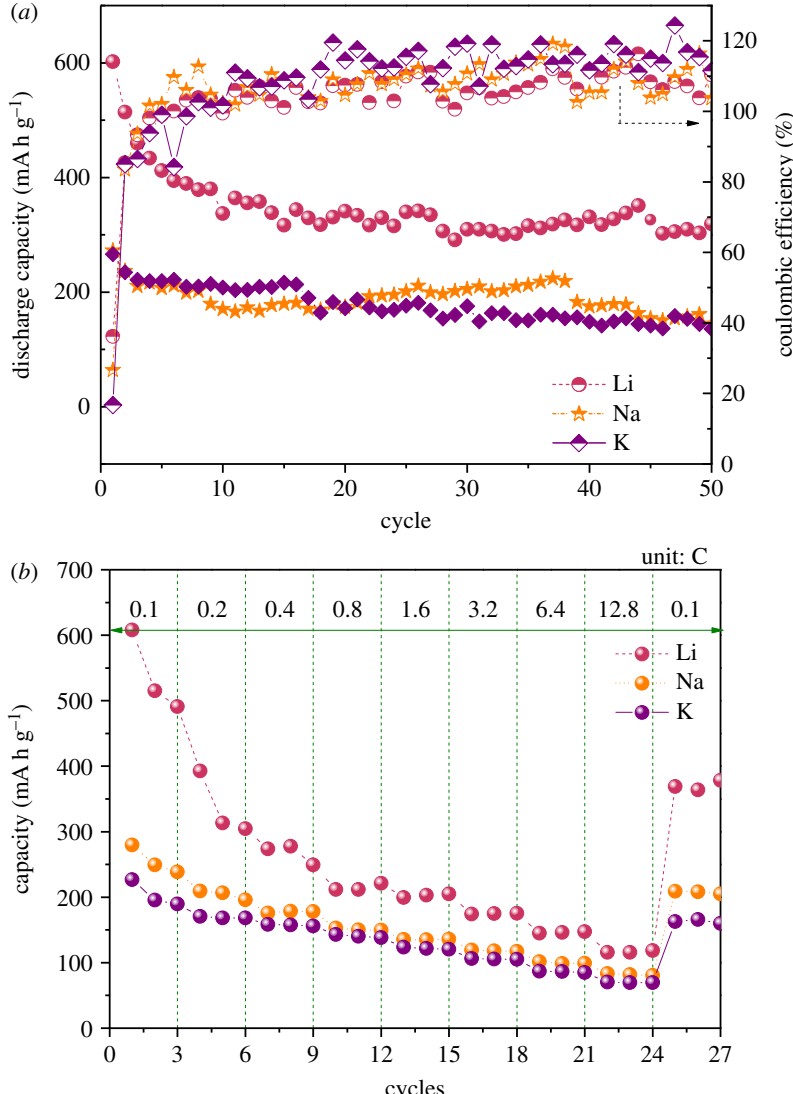

**Figure 7.** Electrochemical characteristics of rGO-250 in Li, Na and K half-cells, respectively, within the 0.02 – 2.0 V potential window. (*a*) Cycle life and (*b*) C-rate performance.

drying in air at various temperatures. As expected, the drying temperatures clearly influence the porous morphology and the electrochemical properties of the prepared rGO samples as anodes in rechargeable battery applications. Specifically, this study demonstrated that the drying temperature of 250°C promoted the formation of optimum pore sizes in the mesopore range and the corresponding rGO material has potential as an anode for emerging rechargeable battery applications, especially in sodium-ion batteries and potassium-ion batteries. Moreover, the synthesis method adopted in the present study offers the advantage of graphene oxide reduction by a simple process of drying with no further chemical/thermal reduction processes. Thus, the present process is a green method for rGO synthesis compared to the usually known chemical reduction process using hydrazine hydrate, sodium borohydride or potassium carbonate. Furthermore, the simple and low-cost synthetic process can be easily upgraded for large-scale production of graphene oxide electrodes for useful energy storage applications provided that simple strategies are used for achieving further exfoliation in the rGO materials to improve their electrochemical properties.

# 4. Conclusion

In this study, we have successfully synthesized rGO samples by modified Hummer's method under ambient atmosphere using appropriate drying temperatures. The rGO samples synthesized by this

simple method are activated to different degrees with increasing drying temperature. The rGO sample dried at the optimum temperature of 250°C demonstrated improved lithium-ion, sodium-ion and potassium-ion storage properties due to the increased specific surface area and porosity, as evidenced by SEM and BET analyses. Furthermore, evaluation of the electrochemical properties using various charge-transport ions confirmed that the electrodes showed decent cyclability and good electrochemical performance at high current densities. The slight instability during repeated cycling that can be related to layer aggregation in the rGO samples calls for improvement in their electrochemical performance versus sodium and potassium, especially. Simple chemical strategies using surfactant/solvent-assisted ultrasonication or mechanical or thermal methods could also be adopted to improve the electrochemical properties in these rGO samples. However, the present experimental results will be useful for improving the full cell characteristics, and especially for preventing the potential drop in sodium-ion batteries and potassium-ion batteries, which are expected to replace the LIB system.

Data accessibility. All the data for this work has been included in the manuscript.

Authors' contributions. J.J., S.L. and J.K. designed and coordinated the study and drafted the manuscript; J.G. and V.M. carried out sequence alignments and helped draft the manuscript; J.S., S.K., M.H.A. and J.L. participated in data analysis; S.K. collected the data. All authors gave final approval for publication.

Competing interests. We declare that we have no competing interests.

Funding. This work was supported by the National Research Foundation of Korea (NRF) grant funded by the Korea government (MSIT) (NRF-2017R1A2A1A17069397). This work was also supported by the Korea Institute of Industrial Technology, as 'Development of the flexible luminescence-battery sheet based on high oxygen/moisture barrier films for the luminescent smart packaging (KITECH JA-18-0040)'.

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
