## [Reviewer comments · Royal Society Open Science]

Review History

RSOS-181978.R0 (Original submission)

Review form: Reviewer 1

Is the manuscript scientifically sound in its present form?

Yes

Are the interpretations and conclusions justified by the results?

Yes

Is the language acceptable?

Yes

Is it clear how to access all supporting data?

Yes

Do you have any ethical concerns with this paper?

No

Have you any concerns about statistical analyses in this paper?

No

Recommendation?

Accept with minor revision (please list in comments)

Comments to the Author(s)

The authors disclosed the preparation of rGO by modified Hummer's method and discussed the influences of the drying temperature to the properties of rGO product. The morphology, BET and C/O ratio of the material have been investigated. Though these points have been discussed and written comprehensively in the manuscript, there are still some issues requiring to be considered.

1) rGO in this paper was synthesized by a modified Hummer's method. Which is the difference between the process used in this paper and previous reports. Is there any difference of the rGO properties compared to that of published before?

2) Table 1 listed the element content of as-synthesized products. Why the oxygen content increased from 22.81% of rGO-250 to 24.46% of rGO-300? Pleas discussed in detail.

3) As shown in the SEM images from Figure 1, the rGO aggregated very much. This would effect the performance of the electrochemical properties. Is there any method to separate the layers of the rGO and obtain the fine dispersion?

4) After 50 cycles, the reversible specific capacity of the half-cell remained at 53%. This is quite poor! Why did the cycle stability deteriorate so quickly?

5) The author should state the novelty of this work more clearly, such as process of preparation, properties of product, or electrochemical performance.

Review form: Reviewer 2

Is the manuscript scientifically sound in its present form?

Yes

Are the interpretations and conclusions justified by the results?

Yes

Is the language acceptable?

Yes

Is it clear how to access all supporting data?

Yes

Do you have any ethical concerns with this paper?

No

Have you any concerns about statistical analyses in this paper?

No

Recommendation?

Major revision is needed (please make suggestions in comments)

Comments to the Author(s)

This manuscript shows a facile synthesis of reduced graphene oxide by modified Hummer's method as anode materials. It seems interesting and there is a lot of work. But this manuscript still needs to be revised before it can be accepted.

1. In the manuscript, there are few literatures in the latest three years (from 2017). I used "rGO" and "battery" to search in Web of Science, I found lots of related literatures. Such as: APPLIED SURFACE SCIENCE, 2019, 465: 470-477
It would be better if the author can add more latest related literatures.
2. It would be better, if the authors can put four figures in Fig. 5 together to see the difference of different samples.
3. It would be better, if the authors can compare their BET result and anode capacity with other literatures.
4. For the electrochemical performance, the authors should provide the CV curve and the Nyquist plots.
5. It seems that, the performance in this manuscript is not good enough.
 - a) The author should compare the performance of the anode using the Hummer's method.
 - b) Why only one temperature was chosen to do sensitivity?
 - c) Please give some comment on how to further improve the performance?

Decision letter (RSOS-181978.R0)

15-Feb-2019

Dear Professor Kim:

Title: Facile synthesis of reduced Graphene oxide by modified Hummer's Method as anode material for Li, Na, and K-ion secondary batteries
Manuscript ID: RSOS-181978

The editor assigned to your manuscript has now received comments from reviewers. We would like you to revise your paper in accordance with the referee and Subject Editor suggestions which can be found below (not including confidential reports to the Editor). Please note this decision does not guarantee eventual acceptance.

Please submit your revised paper before 10-Mar-2019. Please note that the revision deadline will expire at 00.00am on this date. If we do not hear from you within this time then it will be assumed that the paper has been withdrawn. In exceptional circumstances, extensions may be possible if agreed with the Editorial Office in advance. We do not allow multiple rounds of revision so we urge you to make every effort to fully address all of the comments at this stage. If deemed necessary by the Editors, your manuscript will be sent back to one or more of the original reviewers for assessment. If the original reviewers are not available we may invite new reviewers.

To revise your manuscript, log into <http://mc.manuscriptcentral.com/rsos> and enter your Author Centre, where you will find your manuscript title listed under "Manuscripts with Decisions." Under "Actions," click on "Create a Revision." Your manuscript number has been

appended to denote a revision. Revise your manuscript and upload a new version through your Author Centre.

On behalf of the Subject Editor Professor Anthony Stace and the Associate Editor Professor Claire Carmalt.

RSC Associate Editor:
Comments to the Author:
(There are no comments.)

RSC Subject Editor:
Comments to the Author:
(There are no comments.)

Reviewers' Comments to Author:
Reviewer: 1

Comments to the Author(s)

The authors disclosed the preparation of rGO by modified Hummer's method and discussed the influences of the drying temperature to the properties of rGO product. The morphology, BET and C/O ratio of the material have been investigated. Though these points have been discussed and written comprehensively in the manuscript, there are still some issues requiring to be considered.

- 1) rGO in this paper was synthesized by a modified Hummer's method. Which is the difference between the process used in this paper and previous reports. Is there any difference of the rGO properties compared to that of published before?
- 2) Table 1 listed the element content of as-synthesized products. Why the oxygen content increased from 22.81% of rGO-250 to 24.46% of rGO-300? Pleas discussed in detail.

- 3) As shown in the SEM images from Figure 1, the rGO aggregated very much. This would effect the performance of the electrochemical properties. Is there any method to separate the layers of the rGO and obtain the fine dispersion?
- 4) After 50 cycles, the reversible specific capacity of the half-cell remained at 53%. This is quite poor! Why did the cycle stability deteriorate so quickly?
- 5) The author should state the novelty of this work more clearly, such as process of preparation, properties of product, or electrochemical performance.

Reviewer: 2

Comments to the Author(s)

This manuscript shows a facile synthesis of reduced graphene oxide by modified Hummer's method as anode materials. It seems interesting and there is lot of work. But this manuscript still needs to be revised before it can be accepted.

1. In the manuscript, there are few literatures in latest three year (from 2017). I used "rGO" and "battery" to search in Web of Science, I found lots of relate literatures. Such as:
APPLIED SURFACE SCIENCE, 2019, 465: 470-477
It would be better if the author can add more latest relate literatures.
2. It would be better, if the authors can put four figures in Fig. 5 together to see the difference of different samples.
3. It would be better, if the authors can compare their BET result and anode capacity with other literatures.
4. For the electrochemical performance, the authors should provide the CV curve and the Nyquist plots.
5. It seems that, the performance in this manuscript is not good enough.
 - a) The author should compare the performance of the anode using the Hummer's method.
 - b) Why only temperature was chosen to do sensitivity?
 - c) Please give some comment on how to further improve the performance?

Author's Response to Decision Letter for (RSOS-181978.R0)

See Appendix A.

Decision letter (RSOS-181978.R1)

26-Mar-2019

Dear Professor Kim:

Title: Facile synthesis of reduced Graphene oxide by modified Hummer's Method as anode material for Li, Na, and K-ion secondary batteries
Manuscript ID: RSOS-181978.R1

It is a pleasure to accept your manuscript in its current form for publication in Royal Society Open Science. The chemistry content of Royal Society Open Science is published in collaboration with the Royal Society of Chemistry.

On behalf of the Subject Editor Professor Anthony Stace and the Associate Editor Professor Claire Carmalt.

RSC Associate Editor
Comments to the Author:
The authors have thoroughly addressed the reviewers comments and the manuscript can be accepted as is.

Reviewer(s)' Comments to Author:

Appendix A

Dear Dr. Laura Smith,

9th March 2019

Ref: Decision letter for our manuscript entitled “**Facile synthesis of reduced Graphene oxide by modified Hummer’s Method as anode material for Li, Na, and K-ion secondary batteries**” (No. RSOS-181978) dated 15thFeb 2019

Sub: Revision of the above said manuscript after addressing all the points raised by the reviewers and performing changes in the manuscript – reg.

Following your decision letter on the above said manuscript submitted for publication, we are sending the rebuttal letter explaining in detail the questions/issues raised by the reviewers and the changes performed on the manuscript, as per your suggestion. Firstly, we are thankful to the reviewers for providing a favorable response provided for the publication of our manuscript in this esteemed journal. More importantly, we are sincerely thankful to each of the reviewers for rising up important issues in our manuscript to scale up the quality of the article. We wish to state that we have carefully gone through every comment/issue raised and have taken sincere efforts to incorporate the suggestions of the reviewers. We found the comments very helpful and constructive and thank the reviewers for their useful suggestions. We have addressed all the changes recommended by the reviewers and we are confident that the new version of the modified manuscript is easier to understand and has a more fluent and clear scientific discourse. We have noted your comment and according to your request we have clearly gone through the entire comments from the reviewer and addressed these issues point by point and are providing the rebuttal and the revised manuscript for your kind consideration. The revisions, starting with the last submission, are addressed below.

Reviewer: 1

Comments to the Author(s)

The authors disclosed the preparation of rGO by modified Hummer's method and discussed the influences of the drying temperature to the properties of rGO product. The morphology, BET and C/O ratio of the material have been investigated. Though these points have been discussed and written comprehensively in the manuscript, there are still some issues requiring to be considered.

Comment 1: rGO in this paper was synthesized by a modified Hummer's method. Which is the difference between the process used in this paper and previous reports. Is there any difference of the rGO properties compared to that of published before?

The authors wish to thank the reviewer for the valuable time, favorable and useful comments/suggestions toward the publication of this manuscript. The Hummer's method is usually followed to prepare reduced graphite oxide. During the final washing/filtering with water, the solvent molecules tend to occupy the interlayer galleries of the hydrophilic graphite oxide. Upon subsequent annealing of the as-prepared product, the trapped molecules facilitate the formation of the porous morphology in the final graphite oxide powder. Since the porous features of the graphite oxide powder influences their physico-chemical properties, researchers have used various solvents like HBr, NH₃ with different vapor pressures to control the porous morphology and thereby enhance their electrochemical properties. A very recent report indicated that the use of HCl with high vapor pressure (than water) as the final filtering solvent aid in enhancing their porous morphology and simultaneously facilitating partial reduction during the subsequent drying process at only 120 °C. This study demonstrated the enhanced electrochemical properties with respect to rechargeable lithium batteries.

Inspired by the earlier work on HCl, the present work performed a systematic study on using concentrated HCl as the filtering solvent to prepare the reduced graphene oxide (rGO) at various drying temperatures of 120, 200, 250, and 300 °C. The variation of the drying temperatures on the porous morphology and electrochemical properties in the rGO material was studied in

detail for rechargeable lithium, sodium and potassium battery applications. As expected, our work confirmed that the drying temperatures also significantly influenced the surface and electrochemical properties. All these statements have been included in the revised manuscript. The authors are sincerely thankful to the reviewer for raising up this valuable point as the motivation and the uniqueness of the present work could be stated more clearly.

Comment 2: Table 1 listed the element content of as-synthesized products. Why the oxygen content increased from 22.81% of rGO-250 to 24.46% of rGO-300? Pleas discussed in detail.

The authors thank the reviewer for the comment. The authors wish to state that, in addition to drying temperature, annealing environment also can influence the thermal reduction of GO. Usually, controlled reaction environments like inert air atmosphere or vacuum conditions are followed for the thermal reduction of GO. In the present case, as the surface functional groups containing oxygen groups are removed at high temperatures in open-air conditions. In general, the absorbed water are evaporated and hydroxyl/carboxyl group are decomposed under 250 °C. Given that the open-air conditions are oxidizing atmospheres, there can be increase in oxygen content via some chemical reactions like the chemisorption of oxygen by the active surface carbon. Also, it is possible that the impurities present in open air could interfere with the sample and cause undesired reactions. Hence, more studies are required and currently underway to identify the exact reasons for the slight increase in the oxygen content at 300 °C. Nevertheless, the authors are sincerely thankful to the reviewer for his careful analysis of our data and providing crucial comments for the authors to dwell upon and increase the quality of the manuscript.

Statements included in the revised manuscript:

“Interestingly, the oxygen content slightly higher for the rGO prepared at 300 °C than that prepared at 250 °C. This can be related to the open-air environments used for the drying process in the present study as the annealing environment can influence the thermal reduction of GO. Usually, controlled reaction environments like inert air atmosphere or vacuum conditions are

followed for the thermal reduction of GO. In the present case, the surface functional groups containing oxygen (absorbed water and hydroxyl/carboxyl group) are decomposed at temperatures under 250 °C.²² Given that the open-air conditions are oxidizing atmospheres, there can be a slight increase in oxygen content via some chemical reactions like the chemisorption of oxygen by the active surface carbon. Also, it is possible that the impurities present in open air could interfere with the sample and cause undesired reactions. Hence, more studies are required to identify the exact reasons for the slight increase in the oxygen content at 300 °C.”

Comment 3: As shown in the SEM images from Figure 1, the rGO aggregated very much. This would effect the performance of the electrochemical properties. Is there any method to separate the layers of the rGO and obtain the fine dispersion?

The authors wish to thank the reviewer for the comment. In this work, we were tried to use ‘ultra-sonication’ before drying at moderate temperatures. However, it appears that this method was less effective to separate the layers of the rGO and obtain the fine dispersion. This can be one of the reasons for the slight cycling instability observed in the prepared samples. Therefore, an addition step towards exfoliating the sheets need to be considered as an immediate direction of research with these materials. Strategies of performing solvents/surfactants-assisted ultra-sonication or mechanical or thermal methods to improve the exfoliation in the rGO materials. These statements have been included in the revised manuscript. However, the authors are sincerely thankful to the reviewer again for the very useful comment since it has helped to not only improve the quality of the manuscript but also help in finding inroads to improve the properties of the prepared materials.

Statements included in the revised manuscript:

“Overall, the SEM images reveal the slight aggregation of the layers in the prepared samples, especially for the lower temperature samples.”

“Although the aspect of gradual specific capacity decrease, in general, under repeated cycling is most likely related to the aggregated layers in the rGO

samples, there is room for further improvement in the electrochemical properties using simple strategies like solvent/surfactant-assisted ultrasonication and/or mechanical/thermal methods.”

Comment 4: After 50 cycles, the reversible specific capacity of the half-cell remained at 53%. This is quite poor! Why did the cycle stability deteriorate so quickly?

The authors wish to thank the reviewer for the comment. The authors agree to the reviewer that the electrochemical property is still subject to improvement. The initial decrease in the specific capacity during cycling is mostly related to the stabilization of the SEI film formed. The SEI layer was formed as a result of the reaction of transport ions (Li^+ , Na^+ , K^+) with residual oxygen containing functional groups on the electrode surface. The probable cause for the decrease in capacity on short-term cycling is probably related to the irreversible ion-insertion. This can be related to the possibility that the reduced graphene oxide layers require to be more separated to facilitate stable insertion. This is an area that needs to be understood and more studies are underway to improve their electrochemical properties as this work is only a part of the major work aimed to find the practical potentiality of these materials in the field of rechargeable batteries. The present work is only an attempt to show the effect of the drying temperatures upon the electrochemical properties of the prepared rGO samples as anodes in rechargeable battery applications. Finally, the discussed points have been included in the revised manuscript. Nevertheless, the authors wish to sincerely thank the reviewer for raising up this issue and encouraging us to dwell deep on the subject of further improving the electrochemical property of the prepared samples.

Statements included in the revised manuscript:

“The general decrease in the specific capacities during the initial few cycles of the present samples can arise from the stabilization of the solid electrolyte interphase (SEI) layer on the electrode surface.⁸ However, the slight instability in the specific capacities during repeated cycling can be related to the slightly aggregated layers in the prepared electrode samples and further investigations towards layer exfoliation in the rGO samples via chemical or mechanical or

thermal methods are required.”

Comment 5: The author should state the novelty of this work more clearly, such as process of preparation, properties of product, or electrochemical performance.

The authors thank the reviewer for the valuable comment. In this work, we have synthesized reduced graphene oxide (rGO) by a modified Hummer's method using HCl with higher vapor pressure (than water) as the filtration solvent followed by subsequent drying in air at various temperatures. The influence of the drying temperatures upon the porous morphology and the electrochemical properties of the prepared rGO samples as anode materials have been investigated in detail for lithium, sodium and potassium battery applications. This study clearly showed that the drying temperature of 250 °C promoted the formation of optimum pore-sizes in the mesopore range and the corresponding rGO material has potential as anode for emerging rechargeable battery applications, especially, in NIBs and KIBs. Moreover, the presentation of a simple drying process with no further chemical/thermal reduction processes to enhance the porous morphology and improve the electrochemical properties in graphene oxide electrodes for useful energy storage applications is promising. These points have been included in the revised manuscript

The authors are thankful again for the comment as it enabled the description of the work in detail with the purpose and the achievement and thereby improve the readability of the manuscript.

Statements included in the revised manuscript:

“Moreover, recently, the solvent molecules from the water washing/filtration step in the Hummer's method was identified to remain in the interlayers of the hydrophilic as-prepared product. Upon subsequent annealing, the trapped molecules facilitate the formation of the porous morphology in the final product. Since porosity features are known to influence electrochemical properties, various solvents (other than water) like HBr, NH₃ and recently, HCl with different vapour pressures were utilized to tune the porous formation of GO and

thereby enhance their electrochemical performances. Specifically, the use of HCl with high vapor pressure (than water) enabled to obtain porous morphology and simultaneously promote partial reduction in the GO material during drying at 120 °C for anode application in rechargeable lithium batteries.

Inspired by these works, the present work performed a systematic study on using concentrated HCl as the filtering solvent in a modified Hummer's method¹³ to prepare reduced graphene oxide (rGO) at various drying temperatures of 120, 200, 250, and 300 °C. No further chemical/thermal reduction procedure is followed. Electron microscopy and surface analyses confirmed the accordion-morphology of rGO and their mesoporous characteristics. The variation of the drying temperatures on the porous morphology and electrochemical properties in the rGO material was studied in detail. In other words, the feasibility of using the prepared rGO host for the insertion/de-insertion of various carrier ions (such as Li⁺, Na⁺, and K⁺) for energy storage applications is demonstrated. As expected, our work confirmed that the drying temperatures also significantly influenced the surface and electrochemical properties. The present study thus showcases the possibility of using rGO as an electrode material for alternative energy storage systems.”

Reviewer: 2

Comments to the Author(s)

This manuscript shows a facile synthesis of reduced graphene oxide by modified Hummer's method as anode materials. It seems interesting and there is lot of work. But this manuscript still needs to be revised before it can be accepted.

Comment 1: In the manuscript, there are few literatures in latest three year (from 2017). I used “rGO” and “battery” to search in Web of Science, I found lots of relate literatures. Such as:

APPLIED SURFACE SCIENCE, 2019, 465: 470-477. It would be better if the author can add more latest relate literatures.

Firstly, the authors wish to thank the reviewer for the favorable comments and suggestions toward the publications of the manuscript. In accordance to the reviewer's suggestion, we have included a few references in

the Introduction section. The suggested/above said reference has also been included in the revised manuscript. The authors are thankful to the reviewer for suggestion of adequate literature and providing a reference to be cited too.

Comment 2: It would be better, if the authors can put four figures in Fig. 5 together to see the difference of different samples.

The authors are thankful to the reviewer for the comment. In accordance, the figure was merged in the revised manuscript.

Comment 3: It would be better, if the authors can compare their BET result and anode capacity with other literatures.

The authors thank the reviewer for the comment. In agreement to the reviewer's suggestion, the BET results of a few reduced graphene oxide prepared for various applications have been mentioned in the revised manuscript. In addition, one specific result on the rGO used as anode for sodium battery has been included in the revised manuscript. These points have been included in the manuscript. Overall, the authors thank the reviewer for the very useful suggestion and giving us a clear discourse on preparing for the high quality readership of the esteemed journal.

Statements included in the revised manuscript:

“The maximum surface area values obtained here are competitive to those reported for the case of reduced graphene oxide prepared for various applications.³²⁻³⁵”

“For example, Wang et al.,³⁵ developed porous reduced graphene oxide with high surface area ($\sim 330 \text{ m}^2 \text{ g}^{-1}$) and the electrochemical measurement revealed that reversible sodium storage capacities of 174 mAh g^{-1} at 0.2 C. Although the surface area is higher than that measured in the present case, the reversible specific capacity attained here is quite competitive with the value reported.”

Comment 4. For the electrochemical performance, the authors should provide the CV curve and the Nyquist plots.

The authors thank the reviewer for the comment. In accordance, the CV curves for three samples have been provided in the Supporting Information of the revised manuscript. As expected, the higher the drying temperature, greater was the charge capacity stored by the electrode sample for the lithium test cells. Also, the Nyquist plots were compared for the test cells, which were fabricated using all the prepared electrode samples, after completing the 1st charge cycle. The results have indicated that although there is no much variation in the charge transfer resistances at the electrode/electrolyte interface of the test cells, the slightly improved ion-diffusion in the bulk of the cell with the 250 °C sample clearly support the electrochemical data presented for the samples. The detailed explanation of the results have been provided in the revised manuscript. The authors are sincerely thankful to the reviewer for raising up such a valuable issue and making the manuscript better towards its publication.

Statements included in the revised manuscript:

“The cyclic voltammetry profiles recorded for the lithium test cells (Fig. S1) clearly support the trend of the voltage profiles showing increased specific capacity for higher drying temperatures.”

“Nevertheless, the reasoning for the slightly better electrochemical performance of the rGO sample prepared at 250 °C, especially for sodium and potassium test cells, was attempted. To clearly understand the influence of the drying temperature toward the electrochemical performance of rGO electrode in coin cell, PEIS measurement was performed for all rGO samples with Na metal anode under a determined voltage range of 0.02-2.0V and 0.5C (187.5 mA g⁻¹) current density. Fig. 7(a) represents the Nyquist plot of the first-charge impedance behavior in the frequency range of 1MHz-10mHz. The compressed semicircle magnitude is steadily enlarged with raising temperatures and ended at 28.3, 38.7 and 48.9 ohm for the rGO- 120, 200 and 300 samples, respectively. Interestingly, the rGO-250 sample exhibited an out-of-trend behavior thereby suggesting a different mechanism. Hence, an electrical equivalent circuit (EEC) analysis was performed using Z-VIEW software, providing a closet analog description of the electrochemical kinetics. Accordingly, the EEC contained a resistor R_s in series with two parallel RC combinations ($R_{SEI} + CPE1$, $R_{CT} + CPE2$), and connected to a Warburg element, $W_{Diffusion}$, as shown in Fig. 7 (a) (inset). The detailed information of these components is listed in Table 3.

According to our previous study, the physical meaning of the EEC elements that describe the steps for lithium-ion insertion are proposed as follows: (i) R_s is attributed to the ion transport inside the separator and the electrolyte; (ii) the $R_{SEI} + CPE1$ combination represents lithium migration through the SEI layer; (iii) the $R_{CT} + CPE2$ group reflects the charge-transfer at the electrode/electrolyte interface; and (iv) $W_{Diffusion}$ illustrates the lithium-ion diffusion inside the bulk phase of the active material.^{39,40} Taking a cue from these, the variations of R_s , R_{SEI} , R_{CT} , and $R_{Diffusion}$ values at the first-charge state regarding different temperature samples for the sodium-ion test cells are provided in Fig. 7 (b). The system (R_s) and SEI layer (R_{SEI}) resistances occupy a minor distribution in the total resistance, which steadily retained for all samples. Furthermore, the charge-transfer resistance (R_{CT}) tends to increase at the higher temperature of 200 and 300°C samples. Most importantly, the diffusion resistance ($R_{Diffusion}$) of rGO-250 sample shows the lowest values of 16.8 ohm more than three-fold less than rGO-300 samples (56.49 ohm). This implies a drastic improvement in the ion-diffusion conductivity due to the optimized drying temperature of 250°C. This PEIS study matches well with the galvanostatic results in previous sections. Additionally, the best performance is related to their high specific surface area and optimum pore-size distribution, as indicated in Table 2.”

Comment 5: It seems that, the performance in this manuscript is not good enough.

a) The author should compare the performance of the anode using the Hummer’s method.

The authors thank the reviewer for the comment. The authors agree that comparing the performances of the samples prepared from the original Hummer’s method is understandable. However, the sample prepared by this method is not sufficient to enhance graphite oxide property generally. Since this result is well-known, the authors have avoided the inclusion of this sample in the manuscript. Instead, we have attempted to make a meaningful comparison of the electrochemical properties of the prepared samples with the commercially available graphite powder.

b) Why only temperature was chosen to do sensitivity?

Inspired by the previous work of using HCl as a filtration solvent for the preparation of rGO samples in a modified Hummer's reaction, the present work performed a systematic study on using concentrated HCl as the filtering solvent to prepare the reduced graphene oxide (rGO) at various drying temperatures of 120, 200, 250, and 300 °C. The variation of the drying temperatures on the porous morphology and electrochemical properties in the rGO material was studied in detail for rechargeable lithium, sodium and potassium battery applications. As expected, our work confirmed that the drying temperatures also significantly influenced the surface and electrochemical properties. Further studies are underway to understand the other variation parameters and will be studied in detail in the near future. However, the authors are sincerely thankful to the reviewer again for the very useful comment as it will help in improving further the properties of the prepared materials.

c) Please give some comment on how to further improve the performance?

The authors are thankful to the reviewer for the comment. The authors agree to the reviewer that the electrochemical property is still subject to improvement. The performance instability can be related to the observation of slightly aggregated rGO layers. It could be that the method of ultra-sonication used was not very effective to completely separate the layers of the rGO and obtain the fine dispersion. This can be one of the reasons for the slight cycling instability observed in the prepared samples. Therefore, an addition step towards exfoliating the sheets need to be considered as an immediate direction of research with these materials. Strategies of performing solvents/surfactants-assisted ultra-sonication or mechanical or thermal methods to improve the exfoliation in the rGO materials. All these statements have been included in the revised manuscript. The authors wish to sincerely thank the reviewer again for the valuable comment to tune-up the article quality for better readership.

Statements included in the revised manuscript:

“Although the aspect of gradual specific capacity decrease, in general, under repeated cycling is most likely related to the aggregated layers in the rGO samples, there is room for further improvement in the electrochemical properties using simple strategies like solvent/surfactant-assisted ultra-sonication and/or mechanical/thermal methods. In addition, the possibility of

tuning other parameters including the annealing environment and heating rate could also be adopted to arrive at enhanced electrochemical properties.”

In conclusion, we wish to state that all these changes performed are highlighted in yellow in the re-submitted manuscript for easy identification. Also, we have included all the above references in the revised manuscript. We strongly believe that we have considered the recommendation of the reviewers and have performed the changes required in the manuscript and thereby enhanced the manuscript by modifying some of the original sentences and including new sentences to provide an adequate literature review. We are hugely indebted to the reviewers for their patience in providing us very valuable comments towards making our manuscript better for publication.

As the corresponding author, any queries regarding our work would be most welcome to be discussed at jaekook@chonnam.ac.kr

Thank you for your patience and consideration

Thanking you,

Yours sincerely

KIM JAEKOOK

Professor

Department of Materials Science and Engineering

Chonnam National University

Gwangju 500- 757.Republic of Korea.

Mobile : +82-10-2367-0010

Office : +82-62-530-1703

Fax : +82-62-530-1699